# Factors associated with virological non-suppression among HIV-positive children receiving antiretroviral therapy at the Joint Clinical Research Centre in Lubowa, Kampala Uganda

**Sarah Nabukeera**[1]*, **Joseph Kagaayi**[1], **Fredrick Edward Makumbi**[1], **Henry Mugerwa**[2], **Joseph K. B. Matovu**[1,3]

**1** Makerere University School of Public Health, Kampala, Uganda, **2** Joint Clinical Research Center, Kampala, Uganda, **3** Busitema University Faculty of Health Sciences, Mbale, Uganda

* nabukeerasarah6@gmail.com

## Abstract

### Background

While the proportion of HIV-positive children (under 15 years) enrolled on antiretroviral therapy (ART) has increased in recent years, up to 60% of children started on ART do not achieve virological suppression. We set out to determine the factors associated with virological non-suppression among children living with HIV receiving ART at a peri-urban HIV care clinic in Kampala, Uganda.

### Method

This was a retrospective cohort study conducted at the pediatric HIV/AIDS clinic at the Joint Clinical Research Centre (JCRC) in Kampala, Uganda. Three hundred (300) HIV-positive children (0–14 years) were randomly selected from existing medical records and data on children's socio-demographic and clinical characteristics (age at ART initiation, WHO clinical staging, and ART-induced side effects) were abstracted using a data abstraction form. Virological non-suppression was defined as a viral load ≥1000 copies/Ml of blood after six months of ART initiation. Incident rate ratios (IRRs) were determined as a measure of association between virological non-suppression and child/patient characteristics. The IRRs were obtained via a modified Poisson regression with corresponding 95% confidence intervals (95%CI). All analyses were done using statistical package, Stata version 15.

### Results

The overall non-suppression rate among HIV-positive children on ART was 23%. Being at WHO clinical stage 4 at ART initiation [adj. IRR 2.74; 95%CI: 1.63, 4.61] and ART-induced side effects [adj. IRR 1.77; 95%CI: 1.06, 2.97] were significantly associated with non-suppression. Older age at ART initiation (age 5–9 years: [adj. IRR 0.42; 95%CI: 0.28, 0.65]; age

**Data Availability Statement:** All relevant data are within the paper and its Supporting Information files.

**Funding:** Routine collection of the data used in this study was supported by the DAAD: German Academic Exchange Service in Uganda. The funders had no role in study design, data collection and analysis, decision to publish, or preparation of the manuscript.

**Competing interests:** The authors declare that they have no competing interests.

**Abbreviations:** ART, Antiretroviral therapy; ARV, Antiretroviral; CI, Confidence Interval; EFV, Efavirenz; HAART, Highly Active Antiretroviral Therapy; HIV, Human Immunodeficiency Virus; IAC, Intensive Adherence Counseling; IRR, Incidence Rate Ratio; JCRC, Joint Clinical Research Centre; LP/vr, Lopinavir/ritonavir; MoH, Ministry of Health; NVP, Nevirapine; TB, Tuberculosis; UNAIDS, The Joint United Nations Programme on HIV/AIDS; VL, Viral Load; WHO, World Health Organization.

10–14 years: [adj. IRR 0.34; 95%CI: 0.18, 0.64] was less likely to be associated with virological non-suppression.

## Conclusion

Nearly a quarter of HIV-positive children on ART had a non-suppressed viral load after six months of treatment. Being at WHO clinical stage 4 at ART initiation and ART-induced side effects were significantly associated with virological non-suppression while older age at ART initiation was protective. Our findings suggest a need for age-specific interventions, particularly those targeting children below five years of age, to improve virological suppression among HIV-positive children receiving ART in this setting.

## Background

In East and Southern Africa out of the 1,400,000 HIV-positive children, only 51% are on antiretroviral therapy (ART) [1]. Although the overall suppression rate is estimated to be 47% [2], the suppression rate among children is not documented. HIV prevention and treatment efforts primarily aim at reducing morbidity and mortality among people living with HIV, but also to reduce the risk of transmission, hence the need to ensure virological suppression to undetectable levels among children on ART. It is recommended that children with initial positive virological test results are initiated on ART immediately and routine viral load monitoring be carried out at 6 and 12 months, then every 12 months if the patient's viral load becomes stable [3].

In low and middle income countries, a viral load (VL)<1000 copies/ml defines treatment success (suppression), a measure of ART efficacy, which also indicates treatment adherence and reduced risk of HIV transmission [3]. Several factors like; low adherence rate [4], WHO clinical stage 4 and TB co-infection have been highlighted to be associated with virological non-suppresion among adults [5]. Likewise, viral load suppression rates among children on ART have been shown to be low [1] and considerably poorer [6,7]. However, the factors associated with virological non-suppression among children (0-14years) receiving ART are not documented.

In Uganda, regular VL monitoring is done at the Central Public Health Laboratory through the districts' laboratory hubs and in case of virological non-suppression, that is; two consecutive viral loads above 1000 copies/ml done at least 3–6 months apart [8], 3 sessions of intensive adherence counseling (IAC) are offered. IAC is offered to the caregiver or to both the child and the caregiver at one-month intervals by mainly counselors although at times by nurses, clinicians and peer-educators [9]. After the third session, a post-IAC VL test is done and if suppressed, the patient continues with usual treatment and repeats VL test after a year [10]. During IAC, barriers to adherence are identified and the possible ways to overcome these barriers are explored by assessing the patient's adherence level, advising on how to improve adherence and further assisting to make, and arrange an adherence plan. Treatment failure and resistance testing among children must be confirmed before switching to second- or third-line regimens respectively [8].

In Uganda, approximately 95,000 children are living with HIV, 47% of them receiving ART, with a non-suppression rate of 60.1%, which is higher than that observed among adults (40.4%) and also far above the national target of 10% [11]. Although reasons for high non-suppression rates have been explored among adult HIV-positive patients, and young age indicated

as a predictor of non-suppression among children [12], there is a dearth of information on the factors that are associated with high virological non-suppression rates among HIV-positive children enrolled on ART. In this study, we set out to determine the factors that affect virological non-supppression among HIV-positive children receiving ART in the pediatric HIV clinic at the Joint Clinical Research Centre (JCRC) to help HIV program managers to design and modify age-specific strategies or policies to improve treatment outcomes.

## Methods

### Study area

This study was conducted at the Joint Clinical Research Centre (JCRC), hereafter referred to as 'the Centre', a medical research institution in Uganda specializing in HIV treatment and management [13]. The Centre focuses on HIV/AIDS research in all age groups, clinical trials of antiretroviral (ARV) drugs and Tuberculosis (TB) treatment among other research areas. The majority of these projects are implemented in collaboration with national, regional and international organizations, international health research institutions, and non-governmental organizations (NGOs) and universities. The Centre operates a pediatric specialized HIV care clinic that attends to both out-patient and in-patient HIV-positive children aged 0–14 years from all parts of the country and runs from Monday to Friday every week. The Centre is the only HIV treatment center in Uganda with the third line treatment regimens. Approximately 1,019 children are receiving HIV care at the Centre with a comprehensive range of services that include; TB management, nutrition support, psychosocial support, outreaches and a wide range of clinical services. However, the Centre has a lower suppression rate and high switch rate to second-line antiretroviral therapy among HIV-positive children than adults [14]. Furthermore, the Centre has a good data management system which would help us get valid study results.

### Study design

The study used a retrospective cohort study design through the reviews of medical records to establish the child/patient factors associated with virological non-suppression among children (0–14 years) living with HIV receiving ART.

### Study population

The study population was HIV-positive children aged 0–14 years receiving ART in the pediatric HIV clinic at the Centre, enrolled between January 2017 and March 2019. Only children who had been on treatment for at least 6 months and had VL test results taken between 6 and 8 months after initiating ART, were considered for this analysis.

### Sample size determination

The sample size for determining the proportion of children living with HIV who had a non-suppressed viral load after six months of receiving ART was estimated using the Kish Leslie (1964) formula, assuming a 95% Confidence Interval, an estimated prevalence of virological non-suppression among HIV-positive children on ART of 60.1% based on the Uganda population-based HIV-impact assessment report 2017 [11], and a margin of error (precision) of 0.05. Based on these assumptions, we estimated that we would need to include 297 children into the analysis, after adjusting for 10% missing data. This sample size was rounded off to 300.

## Sampling procedure

The children included in the analysis were selected from the pediatric HIV database using systematic random sampling. We used clinic records to generate a sampling frame comprising patients' serial numbers that were assigned to children's files at the time of admission into the HIV clinic. With an estimated sample size of 300 and a population size of 1,019, we determined a sampling interview of three (3), suggesting that every third child on the sampling frame was selected to participate in the study. The first participant lied between the first and the third participant and was selected using simple random sampling, ballot method without replacement.

## Data abstraction procedures

A semi-structured data abstraction form was designed by the first author (SN) and used to extract demographic (age at ART initiation–the age of the child at the start of ART in completed years, sex- either female or male), and clinical (TB-co-infection, WHO clinical staging at ART initiation, adherence level, registered ARV-induced side effects and ART regimen) child/patient-related factors from the HIV care database. We employed research assistants who were nurses working at the Centre to help with the data abstraction. We developed a detailed manual of operations and standard operating procedures to guide data collectors on how to abstract the data from the database, with guidance from the first author. The data abstraction tool was pre-tested at the Centre a week prior to the start of the study. During pre-testing, data for 20 HIV-positive children on ART were abstracted from the pediatric HIV database by the Research Assistants, working closely with a Data Manager. These data were assessed for completeness and accuracy by the first author who also validated each question and the possible responses and refined the tool based on the pre-testing feedback received from the Research Assistants, accordingly. A week later, the study team started full-scale data abstraction to obtain the records for 300 HIV-positive children on ART as needed. The abstraction exercise was monitored frequently to ensure that the process was implemented as expected. At the end of each day of data abstraction, the first author reviewed the data collection forms to ensure completeness and accuracy of records. Incomplete forms were returned to the data collectors to repeat the data abstraction exercise until all essential records were retrieved. We further kept track of the data flow process to avoid loss of data forms, assessed the performance of each research assistant and met with research team regularly to discuss progress or any problems that cropped-up.

## Measures

The primary outcome variable for this study was virological non-suppression after six months following initiation of ART, defined as viral load of at least 1000 copies/ml of blood. Age at initiation in completed years was categorized in to 3 sub-categories: (0–4 years, 5–9 years and 10–14 years), sex was categorized as either male or female, TB-co-infection was measured as currently receiving TB treatment or within the past one year, WHO clinical staging at ART initiation included all the stages; i.e. stages I-IV; adherence level was determined by pill-count/self-reports and sub-categorized as 'poor' (40%-84%, 'medium' (85%-94%) or 'high' adherence (95%-100%), based on the 2018 Ministry of Health's consolidated guidelines for the prevention and treatment of HIV and AIDS in Uganda [8]. Registered ARV-induced side effects were measured as; '*yes*'—having registered any side-effect or '*no*'—having not registered any and ART regimen was categorized as Nevirapine (NVP)-based, Efavirenz (EFV)-based or Lopinavir/ritonavir(LPVr)-based regimens.

## Data analysis

Descriptive and inferential statistics were computed using STATA, version 15. Proportions were determined for variables such as sex, WHO clinical stage at initiation, TB co-infection, ART regimen, and ARV-induced side-effects. Mean and standard deviation for age at initiation and adherence level were used to describe the sample. The outcome variable was computed as a binary, YES = Non-suppression status (viral load≥1000copies/ml of blood) or NO = Suppression status (viral load<1000copies/ml of blood). Bivariate analysis was done using simple modified Poisson regression after checking data for normality and other assumptions. Multivariable modified Poisson regression was used, incident rate ratios as the measure of association, at a level of significance of 0.05 and variables significant at p<0.2 were considered for the multivariable analysis. Biological plausibility, logic and other factors associated with virological non-suppression from existing literature were also considered when choosing variables for model building. We used the Akaike Inclusion Criteria to determine the goodness of fit of the model.

## Ethical consideration

Ethical approval for the study was sought from the Makerere University School of Public Health institutional review board and from the research office at the Joint Clinical Research Centre. Consent was sought from the participants and we ensured that they signed consent forms before participating in the study. Trained nurses were used as research assistants and were further trained on the data management protocol to avoid any breach of confidentiality. No personal identifiers were included. The data were not accessible by any other third parties other than the study team.

## Results

### Respondents' socio-demographic and ART-related characteristics

Table 1 shows the socio-demographic and clinical characteristics of the 300 children who were enrolled into this study. Slightly more than half of the children (*n* = 169, 56.3%) were female. The mean age at initiation of ART as 8.1 years SD (3.44); the minimum age was 1 year and the maximum as 14years, most of them (*n* = 145, 48.3%) were aged 5–9 years. The mean adherence level was 95.6% SD (11.9), the lowest was 40% and the highest was 100% and the majority of children (*n* = 262, 87.3%) had an adherence level between 95%-100%. Thirty-three children (*n* = 33, 11.0%) had been co-infected with TB while 8.7% (*n* = 26) had had ART-induced side effects. Approximately thirty-nine percent (*n* = 116) of the children started ART at WHO clinical stage 1, 39.7% (*n* = 119) at stage 2, 15.0% (*n* = 45) at stage 3 and 6.7% (*n* = 20) at stage 4. Majority of the children (*n* = 115, 38.3%) received Efavirenz-based regimen as first line treatment, 89 children (29.7%) received Nevirapine-based regimen and 96 children (32%) received lopinavir/ritonavir-based regimen.

### Virological suppression among HIV-positive children on ART

Table 2 shows the virological suppression status of children receiving ART stratified by socio-demographic and ART characteristics. The proportion of children living with HIV receiving ART who had a non-suppressed viral load after six months of ART at the pediatric HIV/AIDS clinic at JCRC was 23% (*n* = 69). Thirty percent (*n* = 10) of children with TB had a non-suppressed viral load and the proportion of male children with a non-suppressed viral load was slightly higher (23.7%, *n* = 31) compared to that of the female children (22.5%, *n* = 38). Eighty percent (*n* = 16) of children at WHO clinical Stage 4 had a non-suppressed viral load at six

**Table 1. Socio-demographic characteristics of HIV-positive children on ART.**

| Characteristic | Frequency (N) | Percent (%) |
|---|---|---|
| **Sex** | | |
| Male | 131 | 43.7 |
| Female | 169 | 56.3 |
| **TB-status** | | |
| NO | 267 | 89 |
| YES | 33 | 11.0 |
| **WHO-stage at ART initiation** | | |
| Stage 1 | 116 | 38.7 |
| Stage 2 | 119 | 39.7 |
| Stage 3 | 45 | 15.0 |
| Stage 4 | 20 | 6.7 |
| **ART regimen** | | |
| Efavirenz-based | 115 | 38.3 |
| Nevirapine-based | 89 | 29.7 |
| Lopinavir/ritonavir-based | 96 | 32.0 |
| **ART side–effects** | | |
| NO | 274 | 91.3 |
| YES | 26 | 8.7 |
| **Age at ART initiation** | | |
| | Mean 8.1 years SD (3.44) | (Min 1 year- Max 14 years) |
| 0–4 years | 64 | 21.3 |
| 5–9 years | 145 | 48.3 |
| 10–14 years | 91 | 30.3 |
| **ART adherence level** | | |
| | Mean 95.6 SD (11.9) | (Min 40%—Max I00%) |
| Poor (40%-84%) | 31 | 10.3 |
| Medium (85%-94%) | 7 | 2.3 |
| High (95%-100%) | 262 | 87.3 |

months of treatment, a proportion higher than that of the children at stage 3 (33.3%, $n = 15$), stage 2 (13.5%, $n = 16$) and stage 1 (19%, $n = 22$). The proportion of non-suppressed children on Lopinavir -based regimen was 27.1% ($n = 26$), slightly higher than that of children on Nevirapine-based regimen (21.4%, $n = 19$) and Efavirenz-based regimens (21.0%, $n = 24$). Sixty-five percent ($n = 17$) of children with ART-induced side-effects had non-suppressed viral loads, a proportion higher than that reported among children without such side effects (19%, $n = 52$). The proportion of 0–4 year-olds with a non-suppressed viral load was 47.0% ($n = 30$), much higher than that of the 5–9 year-olds (19.3%, $n = 28$) and the 10–14 year-olds (12.1%, $n = 11$). The proportion of HIV-positive children on ART who had non-suppressed viral loads after 6 months of treatment was higher in children with adherence level of 85%-94% (43.0%, $n = 3$) than in those with an adherence level of 40%-84% (25.8%, $n = 8$) and 95%-100% (22.0%, $n = 58$).

## Factors associated with virological non-suppression

Table 3 shows the results from bivariate and multivariable analysis. Results from the bivariate analysis show that having had TB (incident rate ratio (IRR): 0.73; 95%CI: 0.41,1.28), being at WHO clinical stage 3 (IRR 1.76; 95%CI: 1.00,3.08) and stage 4 (IRR 4.22; 95%CI: 2.73,6.52) at

**Table 2. Virological suppression status of children receiving ART stratified by socio-demographic and ART characteristics.**

| Characteristics | Non-suppressed N = 69 (23%) | Suppressed N = 231 (77%) | p-value |
|---|---|---|---|
| **Sex** | | | |
| Male | 31 (23.7) | 100(76.3) | |
| Female | 38 (22.5) | 131(77.5) | 0.81 |
| **TB-status** | | | |
| No | 59 (22.0) | 208 (78.0) | |
| Yes | 10 (30.3) | 23(69.7) | 0.291 |
| **WHO stage at ART initiation** | | | |
| Stage 1 | 22 (19.0) | 94 (81.0) | |
| Stage 2 | 16(13.5) | 103(86.6) | 0.000 |
| Stage 3 | 15(33.3) | 30(66.7) | |
| Stage 4 | 16(80.0) | 4 (20.0) | |
| **ART Regimen** | | | |
| Efavirenz-based | 24(21.0) | 91(79.1) | |
| Nevirapine-based | 19(21.4) | 70 (78.7) | |
| Lopinavir/r based | 26(27.1) | 70 (73.0) | 0.513 |
| **ART-induced side-effects** | | | |
| No | 52 (19.0) | 222 (81.0) | |
| Yes | 17 (65.4) | 9(34.6) | 0.000 |
| **Age at ART initiation** | | | |
| 0–4 years | 30 (47.0) | 34 (53.1) | |
| 5–9 years | 28(19.3) | 117 (80.7) | |
| 10–14 years | 11 (12.1) | 80 (88.0) | 0.000 |
| **ART adherence level** | | | |
| Poor (40%-84%) | 8 (25.8) | 23 (74.2) | |
| Medium (85%-94%) | 3(43.0) | 4(57.1) | |
| High (95%-100%) | 58(22.0) | 204(78.0) | 0.405 |

the start of ART; having ART-induced side effects (IRR 3.45; 95%CI: 2.37, 4.99) and older age at ART initiation (5–9 years: (IRR 0.41; 95%CI: 0.27, 0.63); 10–14 years: (IRR 0.26: 95%CI: 0.14, 0.48) were significantly associated with non-suppression. After adjusting for potential and suspected confounders, being at WHO stage 4 at the time of initiation (IRR 2.74; 95%CI: 1.63, 4.61) and having ART-induced side-effects (IRR 1.77; 95%CI: 1.06, 2.97) remained the only factors that were significantly associated with non-suppression. Compared to children who initiated ART at the age of 0–4 years, those who initiated ART at the age of 5–9 years (IRR 0.42; 95%CI: 0.28, 0.65) or 10–14 years (IRR 0.34; 95%CI: 0.18, 0.64) were significantly less likely to have non-suppressed viral loads. We found no significant difference in non-suppression levels between children with high ART adherence (95%-100%), those with medium ART adherence (85–94%) and those with poor ART adherence (40%-84%). Furthermore, there was no significant association between non-suppression and variables like sex, TB co-infection and ART regimen.

## Discussion

Our study of the factors associated with virological non-suppression among HIV-positive children (0–14 years) receiving ART at the Centre shows that nearly a quarter (23%) of the children had a non-suppressed viral load after six months of enrolment. Being at WHO clinical staging 4 at ART initiation and having had ARV-induced side effects during treatment were

**Table 3. Child/patient factors associated with virological non-suppression among children receiving ART.**

| Factors | Non-suppression status | | Unadjusted IRR (95% CI) | Adjusted IRR (95%CI) |
|---|---|---|---|---|
| | Yes n (%) | No n (%) | | |
| **Sex** | | | | |
| Male | 31(23.7) | 100(76.3) | 1 | 1 |
| Female | 38(22.5) | 131(77.5) | 0.95 [0.63,1.44] | 0.90[0.61,1.32] |
| **TB-status** | | | | |
| No | 59(22.0) | 208(78.0) | 1 | 1 |
| Yes | 10(30.3) | 23(69.7) | 0.73[0.41,1.28]* | 1.05[0.60,1.85] |
| **WHO-stage at ART initiation** | | | | |
| Stage 1 | 22(19.0) | 94(81.0) | 1 | 1 |
| Stage 2 | 16(13.5) | 103(86.6) | 0.71[0.39,1.28] | 0.75[0.41,1.36] |
| Stage 3 | 15(33.3) | 30(66.7) | 1.76[1.00,3.08]* | 1.61[0.94,2.76] |
| Stage 4 | 16(80.0) | 4(20.0) | 4.22[2.73,6.52]*** | 2.74[1.63,4.61]*** |
| **ART regimen** | | | | |
| Efavirenz-based | 24(21.0) | 91(79.1) | 1 | 1 |
| Nevirapine-based | 19(21.4) | 70(78.7) | 1.02[0.59,1.74] | 1.10[0.67,1.80] |
| Lopinavir/r-based | 26(27.1) | 70(73.0) | 1.29[0.79,2.11] | 1.36[0.86,2.16] |
| **ART-induced side-effects** | | | | |
| No | 52(19.0) | 222(81.0) | 1 | 1 |
| Yes | 17(65.4) | 9(34.6) | 3.45[2.37,4.99]*** | 1.77[1.06,2.97]* |
| **Age at ART initiation** | | | | |
| 0–4 years | 30(47.0) | 34(53.1) | 1 | 1 |
| 5–9 years | 28(19.3) | 117(80.7) | 0.41[0.27,0.63]*** | 0.42[0.28,0.65]*** |
| 10–14 years | 11(12.1) | 80(88.0) | 0.26[0.14,0.48]*** | 0.34[0.18,0.64]** |
| **ART adherence level** | | | | |
| Poor (40%-84%) | 8(25.8) | 23 (74.2) | 1 | 1 |
| Medium (85%-94%) | 3(43.0) | 4 (57.1) | 1.66[0.58,4.27] | 2.28[0.80,6.47] |
| High 95%-100%) | 58(22.0) | 204(78.0) | 0.86[0.45,1.637] | 1.06[0.64,1.74] |

*p-value<0.05

**P-value<0.01

***P-value<0.0001.

the factors that were significantly associated with virological non-suppression while older age at ART initiation was protective. Collectively, these findings suggest a need for HIV treatment and care programs to design, implement and evaluate targeted interventions to ensure virological suppression among children living with HIV receiving ART, with a particular focus on those initiating ART before five years of age; those that experience ART-induced side-effects and those initiating ART at WHO clinical stage 4.

The finding that nearly a quarter of the children had a non-suppressed viral load after six months on ART is worrying given that it is much higher than what has been observed in other studies. For instance, a study conducted among HIV-positive children in rural Zambia found that only 11.5% of the children were virologically non-suppressed after six months of treatment [15]. Similarly, another study conducted in Zimbabwe found that 19% of the children were virologically non-suppressed, although the study was done in a hospital that specialized in managing opportunistic infections and managing complicated referrals of HIV patients on second and third line treatment, which could explain the lower proportion of non-suppression among the children [16]. Nevertheless, all these studies were conducted in tertiary referral

HIV treatment centres, and therefore, the higher proportion of children with non-suppressed viral loads after six months of treatment observed in our study indicates the need to enhance the strategies aimed at improving suppression among HIV-positive children especially in lower level health facilities, especially the public ones since attending a public health facility has been shown to be associated with sub-optimal adherence in sub-Saharan Africa and Asia [17]. Further studies also need to be done in public facilities, to provide different context-specific information on virological non-suppression among HIV-positive children on ART.

Being at WHO clinical stage 4 at ART initiation was significantly associated with non-suppression, as has been confirmed in other studies [5]. This is likely because such children have advanced HIV-infection with severe symptoms like a very low CD4 cell count as well as rapid disease progression, with AIDS-defining illnesses like pneumocystis pneumonia, toxoplasmosis and ctyomegalo infections, among others. All these illnesses need to be identified and attended to by clinicians for the children to achieve suppression, since they halt the effectiveness of ART-regimens [18]. Nonetheless, good adherence is necessary to achieve viral load suppression [19]. Thus, special adherence initiatives should be designed and implemented by HIV-care programs for children at WHO clinical stage 4 to ensure suppression.

Similar to a study done in Lesotho which indicated that HIV patients who got ART-related side effects had poor treatment outcomes [20], our study findings show that children who had ART-induced side-effects were more likely to have non-suppressed viral loads after six months of treatment. Literature further highlights ART-related side-effects as a strong predictor of non-adherence [21], since many patients missed ART because of the side-effects [22]. There is need therefore for HIV-treatment and care programs to systematically assess children with side-effects and provide targeted counseling to the children and caregivers, as well as train the health-care providers on the management of side effects to avert non-adherence and improve treatment outcomes.

We found that children who initiated ART at an older age (five years or older) were less likely to have non-suppressed viral loads after six months of treatment. This was further confirmed when we stratified the analysis by age-group: children who initiated ART at 0–4 years had a higher proportion of virological non-suppression (49%), with this proportion decreasing with increasing age (16% among 5–9 year-olds and 15% among 10-14year-olds). The relationship between older age at ART initiation and the low virological non-suppression is in agreement with previous studies that found that children initiating ART at five years or older were more likely to be virally suppressed after six months of treatment than those that initiated ART at a much younger age [23]. Difficulty in swallowing the different ART formulations especially for infants, toddlers and younger children due to poor palatability, pill burden and the complexity of storage of the liquid formulations which affects the treatment potency may explain the low levels of virological suppression among children below five years of age [24]. Pediatric HIV-programs should therefore innovate and use child-friendly ART, palatable, easy-to-swallow, fixed dose combinations and medications that are easy to store, as well as create avenues to support caregivers of children below 5 years on ART to enhance adherence, to ensure virological suppression.

Although it is well known that poor adherence is associated with non-suppression, the proportion of children with non-suppressed viral loads after 6months of treatment was higher among children with adherence level of 85–94%, than in those with an adherence level of 40–84% and 951–100%. However, this was not statistically significant and the frequency counts of <5 could explain this controverting finding. Besides, the adherence was self-reported and this too may have biased the findings.

This clinic-based study had a number of limitations. For instance, we could not rule out selection bias because HIV positive children seeking services at the Centre could have had

certain special characteristics, therefore the results may not be correctly representing the true picture in the general population. Furthermore, secondary data was used to establish the child/patient factors associated with virological non-suppression which is liable to misclassification bias, although this was minimized by clearly defining the study variables. More to that, the quality of HIV treatment and care services may be different from that elsewhere in Uganda especially in the government health centers, and so the results may not depict the true picture of the circumstances around virological non-suppression among HIV-positive children (0–14 years) receiving ART in Uganda. This could bias the results towards the null hypothesis and therefore we are more likely to see no associations between different variables and the non-suppression status. Although it requires more resources (time, and monetary), a prospective study design would have been the best alternative study design to minimize selection bias and provide more clarity on the temporal sequence of the virological suppression and the above-mentioned associated factors. Nevertheless, despite those highlighted limitations, our study is the first one to explore children aged 0-14years in a high-volume HIV treatment and care center, a unique study setting where HIV patients from various parts of the country receive treatment, and thus it provides a distinctive national viewpoint on the age-specific challenges of non-suppression that require enhanced or targeted interventions to optimize treatment outcomes.

## Conclusion

The proportion of HIV-positive children receiving ART with a non-suppressed viral load after six months was 23%. Being at WHO clinical stage 4 and ART-related side effects were significantly associated with virological non-suppression, while being five years or older at initiation was protective. HIV-program managers therefore need to design targeted age-specific treatment and care policies, programs and interventions or to modify the existing ones, in order to improve suppression among HIV-positive children receiving ART, to achieve the 90-90-90 UNAIDS targets by 2030.

## Supporting information

**S1 Dataset. Factors for virological no-suppression dataset.**
(XLSX)

**S1 File. Data abstraction form.**
(DOCX)

## Acknowledgments

### Declarations

We would like to thank the study participants for sparing their time to respond to the data collection tool. We also thank the research committee at the Joint Clinical Research Center for their cooperation and support during data collection. Special thanks also go to study coordinators and research assistants for their invaluable effort in the design and execution of data collection.

## Author Contributions

**Conceptualization:** Sarah Nabukeera, Joseph Kagaayi, Fredrick Edward Makumbi, Henry Mugerwa, Joseph K. B. Matovu.

**Data curation:** Sarah Nabukeera.

**Investigation:** Sarah Nabukeera, Henry Mugerwa.

**Methodology:** Sarah Nabukeera, Joseph Kagaayi, Fredrick Edward Makumbi, Joseph K. B. Matovu.

**Supervision:** Sarah Nabukeera, Henry Mugerwa.

**Validation:** Sarah Nabukeera, Henry Mugerwa, Joseph K. B. Matovu.

**Writing – original draft:** Sarah Nabukeera, Joseph K. B. Matovu.

**Writing – review & editing:** Sarah Nabukeera, Joseph Kagaayi, Joseph K. B. Matovu.

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
