## [Decision Letter · Decision Letter 0]

23 Dec 2020

PONE-D-20-25133

Factors associated with Virological Non-Suppression among HIV-Positive Children Receiving Antiretroviral Therapy at the Joint Clinical Research Center in Lubowa, Kampala Uganda

PLOS ONE

Dear Ms Sarah Nabukeera,

Thank you for submitting your manuscript to PLOS ONE. After careful consideration, we feel that it has merit but does not fully meet PLOS ONE’s publication criteria as it currently stands. Therefore, we invite you to submit a revised version of the manuscript that addresses the points raised during the review process.

We look forward to receiving your revised manuscript.

Kind regards,

Professor Kwasi Torpey, MD PhD MPH

Academic Editor

PLOS ONE

Journal Requirements:

2.) Please include additional information regarding the survey or questionnaire used in the study and ensure that you have provided sufficient details that others could replicate the analyses. For instance, if you developed a questionnaire as part of this study and it is not under a copyright more restrictive than CC-BY, please include a copy, in both the original language and English, as Supporting Information.

3.) You have mentioned that the questionnaire was pre-tested. Please clarify if it was how it was pretested and validated.

4.) We note that you have indicated that data from this study are available upon request. PLOS only allows data to be available upon request if there are legal or ethical restrictions on sharing data publicly. For information on unacceptable data access restrictions, please see http://journals.plos.org/plosone/s/data-availability#loc-unacceptable-data-access-restrictions.

5.) Your ethics statement should only appear in the Methods section of your manuscript. If your ethics statement is written in any section besides the Methods, please delete it from any other section.

Reviewers' comments:

Reviewer's Responses to Questions

**Comments to the Author**

1. Is the manuscript technically sound, and do the data support the conclusions?

Reviewer #1: Yes

Reviewer #2: Yes

2. Has the statistical analysis been performed appropriately and rigorously? 

Reviewer #1: Yes

Reviewer #2: Yes

3. Have the authors made all data underlying the findings in their manuscript fully available?

Reviewer #1: Yes

Reviewer #2: No

4. Is the manuscript presented in an intelligible fashion and written in standard English?

Reviewer #1: Yes

Reviewer #2: Yes

5. Review Comments to the Author

Reviewer #1: Reviewer’s Report

The aim of this research paper was to identify the possible underlying factors associated with virological non-suppression among children living with HIV on ART. This study was based on previous ones done in other centres which showed high rates of virological non-suppression among HIV children on ART. The authors went forward to establish those factors at their pediatric HIV/AIDS clinic. Purportedly, the study results would be based on, to put in place measures that would improve the treatment outcomes of HIV infected children in their health facility.

The study was well designed, had a reasonable number of participants and used a good analysis technique. The tables are easily understood and match the information in the text. The conclusion made is realistic and in keeping with the study findings.

The authors found that the virological non-suppression was at 23%. Being at WHO clinical stage 4 at ART initiation of ARVs and having ARV-induced side effects during treatment were the identified risk factors significantly associated with virological non-suppression.

Basing on the above observations, this paper would be a good basis for further studies on the same topic and worth publishing in an international journal like PLOS ONE.

However, as a retrospective study, it has its own shortcomings. The possible limitations of this research were elaborated on in their discussion chapter and are understandable.

There are minor issues which the authors should clarify on:

1. Being at WHO clinical stage 4 at ART initiation per se is a risk factor for virological non-suppression, however, there could be other underlying confounding factors common in stage 4 HIV disease which need to be highlighted with examples in the discussion. This will help clinicians to identify and manage them so as to improve the treatment outcomes

2. The authors should comment on the findings that “Non-suppression was higher in children with adherence level of 61-80% (28.5%, n=4), than in those with an adherence level of 40-60% (23.5%, n=4) and 81-100% (22.7%, n=61)”. Page 16, 2nd last sentence. It is well known and expected that poor compliance to ARTs is associated with virological non-suppression.

I a nutshell, this is a research paper that is worth publishing in PLOS ONE

References

1. Ravichandra KR et al. Int J Contemp Pediatr. 2017 Sep;4(5):1743-1747 http://www.ijpediatrics.com

Reviewer #2: General comments

This is an important toward in the light of countries efforts towards the attainment of the UNAIDS 90 90 90 or 95 95 95 targets by 2030. In addition, paediatric HIV is heavy burden mainly in sub Saharan Africa. The challenges with diagnosis, and more importantly managing paediatric HIV makes this paper an important one. While many countries have made significant progress in managing HIV among adults with resultant good virological suppression rate, this has proven to be very challenging for the paediatric population.

Suggestions for improvement

1. The manuscript has no page nor line numbering making review difficult

2. There are still some typographical and grammatical errors, kindly read through again and edit

3. Ethical clearance: state the ethical clearance reference numbers

4. In the abstract methods, you use =>, this is not the standard symbol used. Please use ≥

5. In this study you only focus on suppression rate at 6 months. What was the 12 months? Since this was a retrospective data collection, why was this not reported?

6. Can you state that truly all the 300 children had exactly 6mths on ART viral load results available? Or were some taken between 6-9mths etc. as is commonly the case?

7. In the background, second paragraph, you have this long and difficult to follow sentence “Several factors like; low adherence rate [4], WHO clinical stage 4 and TB co-infection have been highlighted to be associated with virological non-suppression among adults [5], and viral load suppression rates among children on ART have been [1] shown to be low and considerably poorer [6],[7].” Try to break this into 2 sentences to improve clarity.

8. In the methods section under measures, you categorised level of adherence as “Poor 40%- 60%, Medium 61%-80% and High 80% -100%)”. What informed this? My main concern is 80% being considered as high adherence, so explain this and let readers understand why this.

9. Results, table 2: please state the actual p-values

10. Discussion, second paragraph; you compare your rate of viral non-suppression to some countries findings without giving any critic or comparison of the study populations. Example, were the ages similar as in this study and any other factors which could have modified their findings compared with this study?

11. What is the best study design which would have better given a true picture of the factors affection suppression rate? You stated some limitations of this study and I was therefore expecting some thoughts on example what a prospective study could contribute better.

6. PLOS authors have the option to publish the peer review history of their article (what does this mean?). If published, this will include your full peer review and any attached files.

Reviewer #1: **Yes: **Stephenson Musiime

Reviewer #2: **Yes: **DORCAS OBIRI-YEBOAH

---

## [Author Response · Author response to Decision Letter 0]

10 Jan 2021

Point-by-Point Response to Academic Editor and Reviewers’ Comments

1. Academic Editor’s comments

S/N Comments Response

1 Please ensure that your manuscript meets PLOS ONE's style requirements, including those for file naming Thank you for the polite reminder, yes we have ensured that the manuscript meets all the PLOS ONE style requirements.

2 Please include additional information regarding the survey or questionnaire used in the study and ensure that you have provided sufficient details that others could replicate the analyses. For instance, if you developed a questionnaire as part of this study and it is not under a copyright more restrictive than CC-BY, please include a copy, in both the original language and English, as Supporting Information. We developed a data abstraction form in English as part of the study, that was used by the research assistants (nurses). We have provided a copy of this form, in English language, as supporting information.

3 You have mentioned that the questionnaire was pre-tested. Please clarify if it was how it was pretested and validated. Yes, we have provided clarity on how the data abstraction tool was pretested and validated under the data extraction procedures on page 6, as;

The data abstraction tool was pre-tested at the Centre a week prior to the start of the study. During pre-testing, data for 20 HIV-positive children on ART were extracted from the pediatric HIV database by the Research Assistants, working closely with a Data Manager. These data were assessed for completeness and accuracy by the first author who also validated each question and the possible responses and refined the tool based on the pre-testing feedback received from the Research Assistants, accordingly. A week later, the study team started full-scale data extraction to obtain the records for 300 HIV-positive children on ART as needed. The extraction exercise was monitored frequently to ensure that the process was implemented as expected.

4 We note that you have indicated that data from this study are available upon request. PLOS only allows data to be available upon request if there are legal or ethical restrictions on sharing data publicly. For information on unacceptable data access restrictions. In your revised cover letter, please address the following prompts:

 a). If there are ethical or legal restrictions on sharing a de-identified data set, please explain them in detail (e.g., data contain potentially identifying or sensitive patient information) and who has imposed them (e.g., an ethics committee). Please also provide contact information for a data access committee, ethics committee, or other institutional body to which data requests may be sent. There are no ethical or legal restrictions on sharing a de-identified data set, and this has been clearly stated in the rebuttal letter.

 b). If there are no restrictions, please upload the minimal anonymized data set necessary to replicate your study findings as either Supporting Information files or to a stable, public repository and provide us with the relevant URLs, DOIs, or accession numbers. The data set necessary to replicate our study findings has been uploaded to the PLOS ONE. 

5 Your ethics statement should only appear in the Methods section of your manuscript. If your ethics statement is written in any section besides the Methods, please delete it from any other section. The ethics statement appears only in the methods section.

2. Reviewer #1’s comments

S/N Comments Response

1 The study was well designed, had a reasonable number of participants and used a good analysis technique. The tables are easily understood and match the information in the text. The conclusion made is realistic and in keeping with the study findings. Basing on the above observations, this paper would be a good basis for further studies on the same topic and worth publishing in an international journal like PLOS ONE. We thank the reviewer for these compliments. We agree and hope that our paper will serve as a basis for further studies on virological non-suppression among HIV-positive children on ART, as we have recommended.

2 Being at WHO clinical stage 4 at ART initiation per se is a risk factor for virological non-suppression, however, there could be other underlying confounding factors common in stage 4 HIV disease which need to be highlighted with examples in the discussion. This will help clinicians to identify and manage them so as to improve the treatment outcomes Thank you for pointing this out, we had missed indicating it. We have now addressed this concern page 13, as:

This is likely because such children have advanced HIV-infection with severe symptoms like a very low CD4 cell count as well as rapid disease progression, with Aids-defining illnesses like pneumocystis pneumonia, toxoplasmosis and ctyomegalo infections among others. All these illnesses need to be identified and attended to by clinicians for the children to achieve suppression, since they e halt the effectiveness of ART-regimens

3 The authors should comment on the findings that “Non-suppression was higher in children with adherence level of 61-80% (28.5%, n=4), than in those with an adherence level of 40-60% (23.5%, n=4) and 81-100% (22.7%, n=61)”. Page 16, 2nd last sentence. It is well known and expected that poor compliance to ARTs is associated with virological non-suppression. We have also addressed this concern and commented on this issue in the discussion section: second last paragraph, on page 15, as; 

Although it is well known that poor adherence is associated with non-suppression, the proportion of children with non-suppressed viral loads after 6months of treatment was higher in children with adherence level of 85-94% than in those with an adherence level of 40-84% and 951-100%. However, this was not statistically significant and the frequency counts of <5 could explain this controverting finding. Besides, the adherence was self-reported and this too may have biased the findings .

3. Reviewer #2’s comments

S/N Comments Responses

1 This is an important toward in the light of countries efforts towards the attainment of the UNAIDS 90 90 90 or 95 95 95 targets by 2030. In addition, pediatric HIV is heavy burden mainly in sub Saharan Africa. The challenges with diagnosis, and more importantly managing pediatric HIV makes this paper an important one. While many countries have made significant progress in managing HIV among adults with resultant good virological suppression rate, this has proven to be very challenging for the pediatric population. Thank you so much for this feedback. We do hope that this paper will provide the basis for addressing pediatric HIV treatment challenges in sub-Saharan Africa in the efforts to achieve the 90-90-90/ 95-95-95 targets. 

2 The manuscript has no page nor line numbering making review difficult. Thank you so much for raising this, it was an inadvertent error. We have page numbered the manuscript.

3 There are still some typographical and grammatical errors, kindly read through again and edit Yes, we also realized as we read through again, that there were some typographical and grammatical errors. We have edited the manuscript, thank you for pointing this out.

4 Ethical clearance: state the ethical clearance reference numbers This study was conducted for the MPH dissertation for the first author (SN). Makerere University School of Public Health’s Institutional Review Board, the body that cleared the research, doesn’t provide reference numbers for MPH student research studies. This explains why we did not include the ethical reference numbers. 

5 In the abstract methods, you use =>, this is not the standard symbol used. Please use ≥ Thanks again for pointing out this error; we have corrected it. See page 7 (data analysis section) and page 2 (Abstract: methods section) for details.

6 In this study you only focus on suppression rate at 6 months. What was the 12 months? Since this was a retrospective data collection, why was this not reported? We recognize that we alluded to the 12 months in the background section, first paragraph on page3, stating; It is recommended that children with initial positive virological test results are initiated on ART immediately and routine viral load monitoring be carried out at 6 and 12 months, then every 12months if the patient’s viral load becomes stable.

In stating so, we wanted to indicate the time points that are recommended for monitoring children on HIV as per the 2018 Uganda Ministry of Health consolidated guidelines for the prevention and treatment of HIV infection. However, we restricted our study to 6 months because it is expected that with effective treatment and good adherence, the patient must be able to achieve suppression after 6 months on treatment. 

7 Can you state that truly all the 300 children had exactly 6mths on ART viral load results available? Or were some taken between 6-9mths etc. as is commonly the case? Not all the 300 children had VL testing at exactly 6 months. We included children in the study based on availability of VL test results between 6-8months of ART initiation. We had not made this clear before, but now, it is clearly stated under the methods section: study population page 5.

8 In the background, second paragraph, you have this long and difficult to follow sentence “Several factors like; low adherence rate [4], WHO clinical stage 4 and TB co-infection have been highlighted to be associated with virological non-suppression among adults [5], and viral load suppression rates among children on ART have been [1] shown to be low and considerably poorer [6], [7].” Try to break this into 2 sentences to improve clarity. Thank you so much for this important observation. This very long sentence has now been broken into two sentences as shown on page 3, as:

Several factors like; low adherence rate [4], WHO clinical stage 4 and TB co-infection have been highlighted to be associated with virological non-suppression among adults [5]. Likewise viral load suppression rates among children on ART have been shown to be low [1] and considerably poorer [6],[7].

9 In the methods section under measures, you categorized level of adherence as “Poor 40%- 60%, Medium 61%-80% and High 80% -100%)”. What informed this? My main concern is 80% being considered as high adherence, so explain this and let readers understand why this. We have gone ahead to re-categorize the levels of adherence, in accordance with the 2018 Uganda Ministry of Health consolidated guidelines for the prevention and treatment of HIV infection, and this is now well-stated under ‘Measures’ within the methods section. This re-categorization has resulted in minor changes within the original findings but these changes did not affect the original interpretation and/or direction of results in any way. These minor changes appear on pages 2, 6, 8,9 ,10 and 11.

10 Results, table 2: please state the actual p-values Thank you for pointing out this, the actual p-values are now stated on page 10

11 Discussion, second paragraph; you compare your rate of viral non-suppression to some countries findings without giving any critic or comparison of the study populations. Example, were the ages similar as in this study and any other factors which could have modified their findings compared with this study? We now realize how important this is, thank you for bringing it to our attention. We have included a comparison of the findings and/or populations studied; see page 12. This is what we have included in the revised paper: 

For instance, a study conducted among HIV-positive children in rural Zambia found that only 11.5% of the children were virologically non-suppressed after six months of treatment [15]. Similarly, another study conducted in Zimbabwe found that 19% of the children were virologically non-suppressed, although the study was done in a hospital that specialized in managing opportunistic infections and managing complicated referrals of HIV patients on second and third line treatment, which could explain the lower proportion of non-suppression among the children. [16]. Nevertheless, all these studies were conducted in tertiary referral HIV treatment centres, and therefore, the higher proportion of children with non-suppressed viral loads after six months of treatment observed in our study indicates the need to enhance the strategies aimed at improving suppression among HIV-positive children especially in lower level health facilities, especially the public ones, since attending a public health facility has been shown to be associated with sub-optimal adherence in sub-Saharan Africa and Asia [17]. 

12 What is the best study design which would have better given a true picture of the factors affecting suppression rate? You stated some limitations of this study and I was therefore expecting some thoughts on example what a prospective study could contribute better. We missed pointing out this yet it is vital. Thank you for raising this concern. We have stated the best study design that would have given a true picture on page 14, as:

Although it requires more resources (time, and monetary), a prospective study design would have been the best alternative to minimize selection bias and provide more clarity on the temporal sequence of the virological suppression and the above-mentioned associated factors.

---

## [Editor Report · Decision Letter 1]

14 Jan 2021

Factors associated with Virological Non-Suppression among HIV-Positive Children Receiving Antiretroviral Therapy at the Joint Clinical Research Center in Lubowa, Kampala Uganda

PONE-D-20-25133R1

Dear Ms Nabukeera,

We’re pleased to inform you that your manuscript has been judged scientifically suitable for publication and will be formally accepted for publication once it meets all outstanding technical requirements.

Kind regards,

Professor Kwasi Torpey, MD PhD MPH

Academic Editor

PLOS ONE

Additional Editor Comments (optional):

Comments addressed
---

## [Editor Report · Acceptance letter]

18 Jan 2021

PONE-D-20-25133R1 

Factors Associated with Virological Non-Suppression among HIV-Positive Children Receiving Antiretroviral Therapy at the Joint Clinical Research Centre in Lubowa, Kampala Uganda 

Dear Dr. Nabukeera:

I'm pleased to inform you that your manuscript has been deemed suitable for publication in PLOS ONE. Congratulations! Your manuscript is now with our production department. 

Kind regards, 

on behalf of

Professor Kwasi Torpey 

Academic Editor

PLOS ONE